# Comparison of Transcatheter Aortic Valve Implantation Devices in Aortic Stenosis: A Network Meta-Analysis of 42,105 Patients

**DOI:** 10.3390/jcm11185299

**Published:** 2022-09-08

**Authors:** Ala Abu Dogosh, Ahlam Adawi, Aref El Nasasra, Carlos Cafri, Orit Barrett, Gal Tsaban, Rami Barashi, Edward Koifman

**Affiliations:** 1Soroka Medical Center, Heart Institute, Ben-Gurion University of the Negev, Beer Sheva 84101, Israel; 2Meir Medical Center, Tel Aviv University, Tel Aviv 6423906, Israel

**Keywords:** aortic valve disease, TAVI, all-cause mortality

## Abstract

*Background:* In recent years, trans-catheter aortic valve implantation (TAVI) has emerged as an excellent alternative to surgical aortic valve replacement (SAVR). Currently, there are several approved devices on the market, yet comparisons among them are scarce. We aimed to compare the various devices via a network meta-analysis. *Methods:* We performed a network meta-analysis including randomized controlled trials (RCTs) and propensity-matched studies that provide comparisons of either a single TAVI with SAVR or two different TAVI devices and report clinical outcomes. *Results:* We included 12 RCT and 13 propensity-matched studies comprising 42,105 patients, among whom 27,134 underwent TAVI using various valve systems (Sapien & Sapien XT, Sapien 3, Corvalve, Evolut & Evolut Pro, Acurate Neo, Portico). The mean follow-up time was 23.4 months. Sapien 3 was superior over SAVR in the reduction of all-cause mortality (OR = 0.53; 95%CrI 0.31–0.91), while no significant difference existed between other devices and SAVR. Aortic regurgitation was more frequent among TAVI devices compared to SAVR. There was no significant difference between the various THVs and SAVR in cardiovascular mortality, myocardial infarction, NYHA class III-IV, and endocarditis. *Conclusions:* Newer generation TAVI devices, especially Sapien 3 and Evolut R/Pro are associated with improved outcomes compared to SAVR and other devices of the older generation.

## 1. Introduction

Aortic valve disease is the third most common cause of cardiovascular disease in the United States (U.S.), affecting an estimated 2.5 million adults [1]. In developed countries, aortic stenosis (AS) is the most prevalent of all valvular heart diseases. The prevalence of the disease rises with age [2], affecting up to nearly 10% of patients over 80 years of age [3]. In the context of present-day medicine, despite these numbers, no effective pharmacological treatment is available [3]. Treatment in AS is based on valve replacement, which can reverse the pathophysiological process and improve survival to the level of control patients [4].

Surgical aortic valve replacement (SAVR) was the gold standard treatment for decades before the development of transcatheter aortic valve implantation (TAVI). In recent years, TAVI has emerged as an attractive, less invasive alternative to SAVR for appropriately selected patients with improved outcomes and faster recovery compared to SAVR [1,3,5,6,7,8,9]. In this significant proportion of patients, TAVI has been shown to be safe and effective [10], making it a widely accepted procedure for the treatment of severe AS patients [11,12].

Different transcatheter valves exist with various mechanisms, such as balloon-expandable (BE) or self-expandable (SE) transcatheter heart valves (THVs) [13]. Regarding this matter, the choice is still controversial because there is scarce data comparing the different transcatheter heart valves in terms of survival and quality of life and with regards to significant complications, including stroke, paravalvular leak, bleeding, acute kidney injury (AKI) and conduction abnormalities [14].

Pairwise comparisons of clinical and hemodynamic outcomes with new transcatheter aortic valve replacement prostheses are needed to help select the appropriate device [15].

In the present study, we compared TAVI devices using a systematic review and a Bayesian network meta-analysis.

## 2. Methods

The primary objective of this network meta-analysis was to compare the various TAVI devices, with a common comparator of SAVR, with regards to clinical outcomes including all-cause mortality, cardiovascular mortality, stroke, bleeding, vascular complications, aortic regurgitation, rehospitalization, reintervention, pacemaker implantation, acute kidney injury (AKI), endocarditis, atrial fibrillation and myocardial infarction. Clinical outcomes and event rates are based on the definitions given and the reported incidents in each study. We included all devices with reported data, including prior generation devices such as balloon expandable Sapien and Sapien XT, Sapien 3, Corevalve, Evolut R and Pro, Accurate Neo and Portico. In order to address the generation difference, we included Sapien and Sapien XT in a single group, as well as first-generation Corevalve in a single group, while Sapien 3 was included in a separate group as well as Evolut R and Pro, which enabled features such as repositionability. In trials and studies in which more than a single generation device was used, we performed the categorization according to the device, with over 50% use in the specific study. Devices that are not commercially available were not included in the current analysis, and studies without outcome reports or lack of matching were not included as well. Three independent investigators (AA and AAD and AEL) had systematically screened (August 2020) MEDLINE/PubMed/Ovid/Embase for titles and abstracts containing the terms “TAVI” OR “TAVR” OR “Aortic stenosis,” reviewed the full-text articles and determined their eligibility. Included in the meta-analysis were RCTs and observational studies comparing at least two of the listed valve replacement options for aortic stenosis with available clinical follow-up separately for each treatment arm. Studies with inadequate outcome data, duplication of data and those available only in abstract form were excluded from the analysis. Data were abstracted in accordance with the Preferred Reporting Items for Systematic Reviews and Meta-Analyses (PRISMA) and Meta-analysis Of Observational Studies in Epidemiology (MOOSE) guidelines [16,17] The type of study, year of publication, time of follow up, treatment allocation and valve replacement strategy, patients’ age, gender, co-morbidities, left ventricular ejection fraction (LVEF) and outcome data for all clinical outcomes at the longest available follow-up were extracted and recorded when available. We accepted the studies’ definitions of adverse events.

### Statistical Analysis

Dichotomous variables are expressed as percentages and continuous variables as mean ± standard deviation or median+ IQR (interquartile range) based on normal distribution. To compare directly and indirectly between the aortic valve replacement modalities SAVR and TAVI (various commercially available valves), we used a mixed treatment comparison model generation performed by GeMTC 0.14.3 software (GeMTC, http://drugis.org/software/r-packages/gemtc, Copyright ©2009-2012 Gert van Valkenhoef, accessed on 30 June 2020). A Bayesian hierarchical random-effects model with a directed acyclic graph model for general-purpose Markov chain Monte Carlo analysis was performed with 50,000 tuning iterations and 100,000 simulation iterations. The data are presented as odds ratios (OR) and 95% credible intervals (CrI). Convergence was appraised graphically according to Gelman and Rubin [18] Data from a consistency model are presented, and the direction of the findings was confirmed with an inconsistency model to serve as a sensitivity analysis. Additional sensitivity analysis was performed with the removal of one study at a time to confirm the directionality and magnitude of the findings. Statistical significance was defined as a *p*-value < 0.05.

Data were abstracted by the students in accordance with the Preferred Reporting Items for Systematic Reviews and Meta-Analyses (PRISMA) AND Meta-analysis of Observational Studies in Epidemiology (MOOSE) guidelines.

## 3. Results

We screened and reviewed a total of 4421 MEDLINE citations using the previously defined search terms. About 70 abstracts that met the inclusion/exclusion criteria were evaluated, and the full-text publications were reviewed in detail. Finally, we entered 25 studies in the meta-analysis, including 12 RCTs and 13 observational studies. The study flow chart is shown in Figure 1.

The characteristics of the studies included in the meta-analysis are presented in Table 1. Among the 42,105 patients with aortic stenosis identified from the included articles, 14,971 underwent SAVR and 27,134 underwent TAVI using various valve systems, as described in Figure 2. The mean follow-up period was 23.4 months. The patients’ baseline characteristics are shown in Table 2. The mean age was 78.5 ± 6.1 years. Men comprised 51.3% of the population, and 31% had diabetes mellitus. Prior MI was present in 11.9%, 26.8% of patients had undergone previous PCI and 16.9% had prior coronary artery bypass graft (CABG) surgery. The mean left ventricular ejection fraction (LVEF) was 54.8 ± 11.2%.

The network plot is presented in Figure 3. The Bayesian network meta-analysis demonstrated superiority of Sapien 3 over SAVR in the reduction of all-cause mortality (OR = 0.53; 95%CrI 0.31–0.91), while no significant difference existed between other devices and SAVR (Figure 4). There was also no significant difference between the various TAVI devices and SAVR in cardiovascular mortality. Stroke was less prevalent among patients treated with Sapien 3 (OR = 0.58 [CrI95% = 1.00–0.33]) while Evolut R/Evolut Pro, Portico and Acurate Neo demonstrated a trend for less stroke compared with SAVR. While all TAVI devices showed higher rates of vascular complications, bleeding was less frequent among all devices, with statistical significance in the Sapien 3 and EvolutR/Evolut Pro groups. AKI was less prevalent in TAVI, with statistical significance in the Evolut R Evolut Pro group (OR = 0.19 [CrI95% 0.99–0.14]). NYHA 3–4 following valve intervention was similar between the different TAVI devices and SAVR; however, rehospitalization was less noted in the newer generation devices such as Sapien 3 (OR=0.12 [CrI95% 0.27-0.05]) and Evolut R/Evolut Pro (OR=0.11 [CrI95% 0.33-0.03]) and reintervention was more frequent among the older Corevalve device compared with SAVR. Aortic regurgitation was more frequent among all TAVI devices compared to SAVR, as well as pacemaker implantation, while atrial fibrillation was less frequent among almost all TAVI devices. MI occurred less frequently among Sapien 3 than SAVR (OR = 0.32 [CrI95% = 0.91–0.11]), and endocarditis rates were similar among all devices and SAVR.

Rankings according to the probability of being the best devices among the various TAVI devices and SAVR based on the Bayesian network meta-analysis revealed that Sapien 3 was ranked as having the best probability for being the most effective valve in reduction of all-cause mortality, cardiovascular mortality, stroke and bleeding, while Evolut R/Evolut Pro was ranked together with Sapien 3 with regards to atrial fibrillation, rehospitalization and AKI, as shown in Figure 5. SAVR and Sapien devices were ranked highest probability for decreased incidence of pacemaker implantation. Sapien 3 and SAVR were ranked as having the highest probability of decreasing the risk of aortic regurgitation.

When limiting the analysis to RCTs, there was no statistically significant difference in all-cause mortality, stroke and aortic regurgitation between the various devices and SAVR; however, the overall trend and ranking analysis yielded similar results. (Figure 6). While heterogeneity and quality differences do exist between studies in specific outcome comparisons (Appendix A), inconsistency and node-split analysis also produced similar results.

## 4. Discussion

The main findings of our network meta-analysis conducted with 25 studies and RCT with 42,105 patients with severe aortic stenosis show the advancement of TAVI devices with improved safety and efficacy, especially with newer generation Sapien 3 and Evolut R/Evolut Pro. The analysis points toward superiority over SAVR in terms of all-cause mortality, which is statistically significant with Sapien 3, along with reduction in other important adverse events such as stroke, AKI and rehospitalization, despite inferiority in terms of residual aortic regurgitation and pacemaker implantations. Waqas et al. reported in a meta-analysis several predictors for pacemaker implantation, such as male sex, baseline atrioventricular conduction delays, intraprocedural atrioventricular block, and the use of mechanically expandable and self-expanding prostheses in patients undergoing TAVI [41].

A previous meta-analysis by Ando et al. [42] comparing TAVI using new- versus early-generation valves (Acurate Neo, Direct Flow, Evolut R, Lotus and Sapien 3 versus CoreValve, Sapien and Sapien XT) reported a lower rate of early ≥ moderate AR but remained similar all-cause mortality and pacemaker implantation among patients with new-generation valves. Another network meta-analysis by Takagi et al. [43] comparing new- versus early-generation valves found a relative advantage for Sapien 3 in reducing all-cause mortality when compared to other valves, whereas Lotus valve was best for reduction of incidence of ≥ moderate AR and Acurate best for decreased incidence of pacemaker implantation. In a more recent meta-analysis of 5 randomized controlled trials, TAVI was associated with reduced all-cause mortality and stroke in patients with low surgical risk compared to SAVR. This benefit was not replicated in patients with intermediate surgical risk [44]. Another meta-analysis of 8 studies (RCTs and observational) from Saleem et al. found no statistically significant difference between TAVI and SAVR in mortality and stroke [45]. In another meta-analysis by Siontis et al. comprising RCT’s, TAVI was associated with a reduction of all-cause mortality compared to SAVR, irrespective of valve system and STS score [46]. In this current meta-analysis, which includes a larger number of patients with commercially available devices, according to rankings probability, Sapien 3 valve was the most effective for decreased incidence of all-cause and cardiovascular mortality, stroke and more than moderate aortic regurgitation.

Needless to say, a Heart Team approach is required for every patient, with a meticulous and careful consideration of the clinical, electrocardiographic and anatomical factors, as each device has it’s advantages and disadvantages; however, in the majority of cases, several types of devices can be implanted with excellent results. Moreover, the development of embolic protection devices and improvement of implantation techniques to reduce pacemaker rate will further advance the TAVI procedure to become safer with less complications [47,48]

The present results should be interpreted with caution because of their limitations. First, the meta-analysis included both RCTs and observational studies, which may have selection biases, as there could be additional confounders that could impact the results and were not necessarily reported. Nonetheless, a sensitivity analysis that included only randomized controlled trials showed an overall similar result in all outcomes. Second, there have been important changes in valve design over the time period of some of the included studies. These changes may not be precisely reflected in this analysis. Third, the duration of follow-up in our meta-analysis was up to 2 years, while longer follow-up will be important to determine long-term outcomes and durability of the valves. Finally, different definition criteria and inconsistent reporting of some outcomes across the trials preclude meta-analysis of other patient subgroups and additional outcomes of interest, such as valve thrombosis, valve gradient, valve area, patient-prothesis mismatch or paravalvular regurgitation.

In conclusion, our meta-analysis demonstrated that newer generation TAVI devices, especially Sapien 3 and Evolut R/Pro might be associated with lower rates of all-cause mortality. Further research is needed to clarify the long-term outcomes and durability of various valve systems.

## Figures and Tables

**Figure 1 jcm-11-05299-f001:**
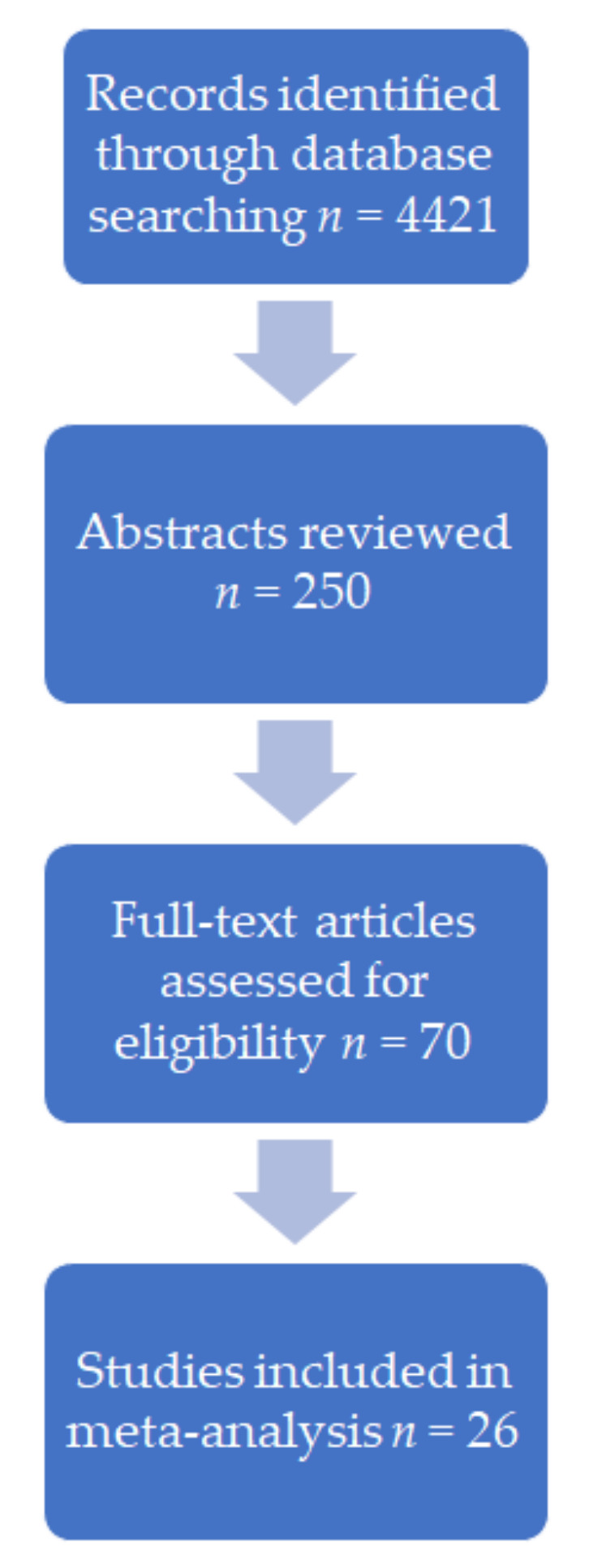
Flow chart showing the process of selecting studies for the meta-analysis.

**Figure 2 jcm-11-05299-f002:**
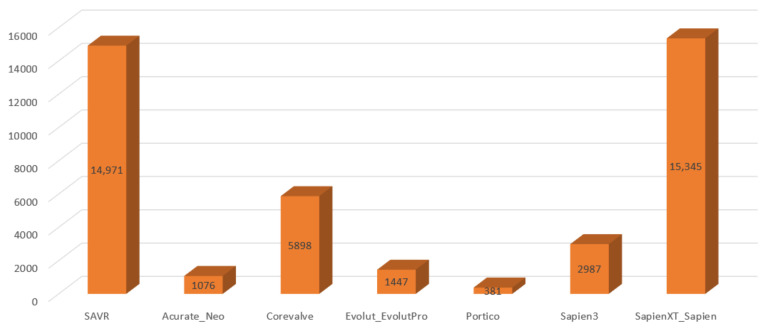
Representativeness of SAVR and various TAVI valves in the included studies. SAVR: surgical aortic valve replacement, TAVI: transcatheter aortic valve implantation.

**Figure 3 jcm-11-05299-f003:**
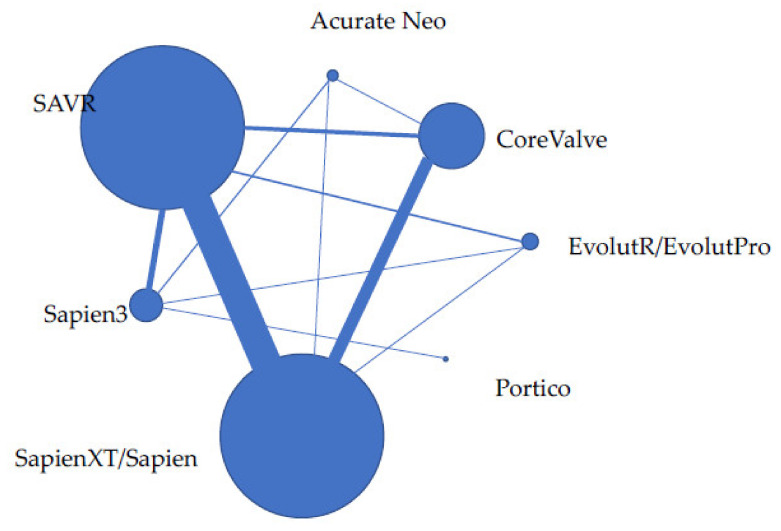
Network diagram of various TAVI valves and SAVR for all-cause mortality. The size of the nodes is proportional to the number of individuals assigned to each valve and the thickness of the lines to the number of direct comparisons in studies.

**Figure 4 jcm-11-05299-f004:**
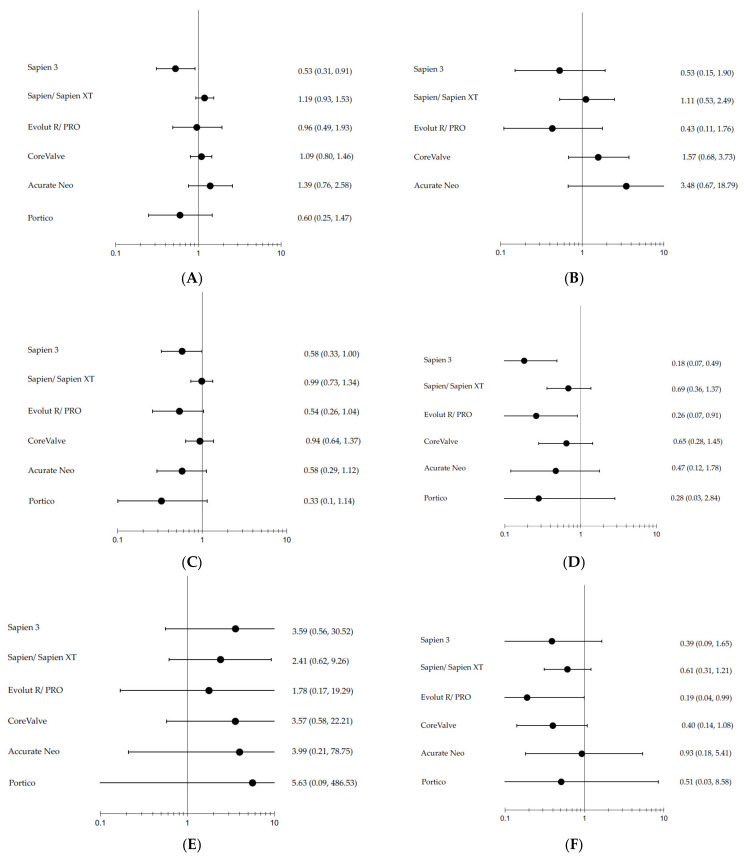
Forest plots of (**A**) all-cause mortality, (**B**) cardiovascular mortality, (**C**) stroke, (**D**) bleeding, (**E**) vascular complications, (**F**) acute kidney injury, (**G**) pacemaker implantation, (**H**) rehospitalization, (**I**) atrial fibrillation, (**J**) aortic regurgitation, (**K**) endocarditis, (**L**) myocardial infarction, (**M**) heart failure NYHA III-IV, and (**N**) reintervention comparing various TAVI valves to SAVR.

**Figure 5 jcm-11-05299-f005:**
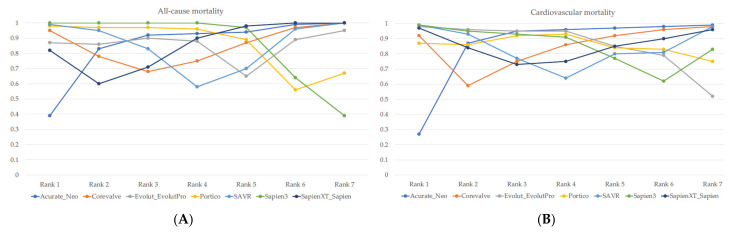
Ranking chart of various TAVI valves and TAVI in (**A**) all-cause mortality, (**B**) cardiovascular mortality, (**C**) stroke, (**D**) bleeding, (**E**) vascular complications, (**F**) acute kidney injury, (**G**) pace-maker implantation, (**H**) rehospitalization, (**I**) atrial fibrillation, (**J**) aortic regurgitation, (**K**) endocarditis, (**L**) myocardial infarction, (**M**) heart failure NYHA III-IV, and (**N**) reintervention.

**Figure 6 jcm-11-05299-f006:**
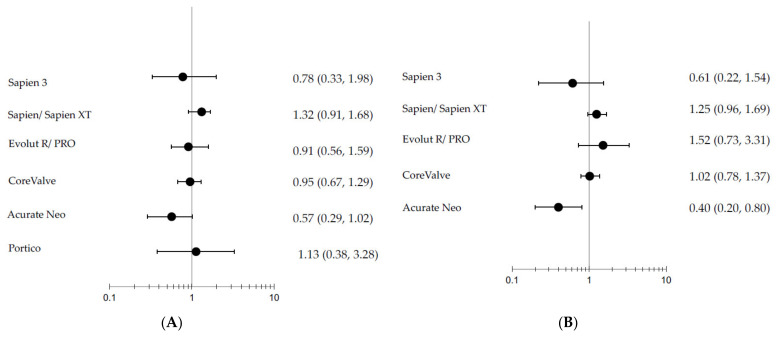
Forest plots of: (**A**) all-cause mortality, (**B**) cardiovascular mortality, (**C**) stroke, and (**D**) aortic regurgitation between various TAVI valves compared to SAVR in randomized controlled trials.

**Table 1 jcm-11-05299-t001:** Study characteristics.

Study	Year of Publication	Follow-Up Time (mo)	Design	Cohort Size (*n*)	Groups (*n*)
Brennan [1]	2017	12	Propensity	9464	Sapien *n* = 4732 SAVR *n* = 4732
Evolut low risk [9]	2019	24	Randomized	1403	EvolutR *n* = 725 SAVR n = 678
Tanush Gupta [12]	2018	24	Propensity	7760	Sapien *n* = 3880 SAVR *n* = 3880
Choice [19]	2020	60	Randomized	241	SapienXT *n* = 121 Corevalve *n* = 120
Corevalve pivotal [20]	2018	60	Randomized	750	Corevalve *n* = 391 SAVR *n* = 359
France propensity [21]	2020	24	Propensity	7820	Corevalve *n* = 3910 SapienXT *n* = 3910
Gerhard Schymik [22]	2015	36	Propensity	432	Sapien + sapienXT *n* = 216 SAVR *n* = 216
Husser [23]	2017	1	Propensity	933	Acurate neo *n* = 311 SAVR *n* = 622
Israeli registry [24]	2019	1	Propensity	735	Sapien3 *n* = 223 EvolutR *n* = 512
LRT [25]	2018	1	Propensity	919	Sapien3 *n* = 200 SAVR *n* = 719
Notion [26]	2019	60	Randomized	280	Corevalve *n* = 145 SAVR *n* = 135
PORTICO IDE [27]	2020	1	Randomized	750	Portico *n* = 381 Sapien3 *n* = 369
Partner I [28]	2015	60	Randomized	699	Sapien *n* = 348 SAVR *n* = 351
Partner II [29]	2020	60	Randomized	2032	SapienXT + 3 *n* = 1011 SAVR *n* = 1021
Partner III [30]	2019	24	Randomized	950	Sapien3 *n* = 496 SAVR *n* = 454
SCOPE II [31]	2020	12	Randomized	796	Acurate neo *n* = 398 Corevalve *n* = 398
Scope I [32]	2019	1	Randomized	739	Acurate neo *n* = 367 Sapien *n* = 364
Solve [33]	2021	12	Randomized	436	Sapien3 *n* = 212 EvolutPRO *n* = 210
SURTAVI [34]	2017	24	Randomized	1574	Corevalve *n* = 864 SAVR *n* = 796
Castordeza [35]	2016	12	Propensity	140	Corevalve *n* = 70 SAVR *n* = 70
Auffret [36]	2017	1	Propensity	321	SapienXT+ Sapien *n* = 122 SAVR *n* = 199
Latib [37]	2012	12	Propensity	222	SapienXT+ Sapien *n* = 111 SAVR *n* = 111
Schaefer [38]	2019	1	Propensity	218	SapienXT+ Sapien *n* = 109 SAVR *n* = 109
Thourani [39]	2016	12	Propensity	2021	Sapien3 *n* = 1077 SAVR *n* = 944
Tzamalis [40]	2020	72	Propensity	407	SapienXT + Sapien *n* = 209 SAVR *n* = 198

**Table 2 jcm-11-05299-t002:** Patient demographics and comorbidities.

Study	Age (Mean ± SD)	Male (%)	Ejection Fraction (Mean ± SD)	Diabetes (%)	Smoking (%)	Hypertension (%)	Dyslipidemia (%)	CABG (%)	PCI (%)	MI (%)
CHOICE	80.7 ± 6.2	35.7	53.7 ± 12.8	29	NA	NA	NA	14.1	39.4	12.45
CoreValve Pivotal	83.3 ± 6.7	52.7	NA	39.7	NA	NA	NA	30.4	35.9	NA
Evolut Low Risk	73.9 ± 5.9	65.1	61.8 ± 7.8	31	NA	83.7	NA	2.3	13.5	5.75
FRANCE propensity	83.5 ± 8	48.9	54.8 ± 14.6	25.7	NA	66.5	NA	11.4	NA	NA
Gerhard Schymik	78.3 ± 4.9	48.8	62.1 ± 10.9	NA	NA	NA	NA	NA	NA	2.75
Husser	81 ± 6	42.8	NA	32.5	NA	NA	NA	9.3	37.7	10
Israeli Registry	82 ± 5	51	NA	40	5.5	84.5	69.5	NA	NA	NA
Brennan	81.5 ± 4.5	52	NA	NA	NA	NA	NA	30.5	26.5	23.3
LRT	71 ± 15.5	60.7	60.9 ± 17.7	25.7	NA	82.58	NA	2.7	12.18	6.85
Notion	79.1 ± 4.8	53.2	NA	19.3	NA	73.6	NA	NA	8.2	5
PorticoIDE	83.3± 7.3	47.3	57.4 ± 11.3	38	NA	NA	NA	21.8	28.6	13
Partner I	84 ± 6.6	57.3	52.9 ± 13.2	NA	NA	NA	NA	43.4	33.3	28.4
Partner II	81.6 ± 6.7	54.5	55.8 ± 11.4	35.9	NA	NA	NA	24.6	27.4	17.9
Partner III	73.5 ± 6	65.8	66 ± 8.8	29.2	NA	NA	NA	NA	NA	5.75
Scope II	83.15 ± 4.3	32.5	NA	28	3.5	85.5	51	5.5	25.5	8.5
Scope I	82.8 ± 4.1	43	56.8 ± 10.9	30.5	2.5	91.5	58	8	32.5	11.63
Solve	81.6 ± 5.5	48.9	NA	33.6	4.1	90.6	40.1	10	37.2	NA
SURTAVI	79.8 ± 6.2	56.4	NA	34.5	NA	NA	NA	16.6	21.3	15.1
Castordeza	78.5 ± 8.2	50	58 ± 13.8	31.4	NA	68.5	NA	NA	NA	NA
Tanush Gupta	77 ± 9.9	78.75	NA	44.6	4.15	84	NA	NA	21	20.75
Auffret	72.4 ± 9.3	35.7	54.6 ± 13.2	NA	NA	NA	NA	12.5	NA	NA
Latib	79.9 ± 7.4	44.1	53.5 ± 12.5	20.2	NA	69.8	NA	NA	NA	14.41
Schaefer	75.15 ± 9.1	50	NA	20.5	NA	NA	NA	NA	NA	4.5
Thourani	81.75 ± 6.7	58.5	57 ± 14.9	NA	NA	NA	NA	27	29.5	17
Tzamalis	78.25 ± 5.2	48.85	62.1 ± 11.4	NA	NA	1.4	NA	NA	NA	2.7

CABG: coronary artery bypass graft, PCI: percutaneous coronary intervention, MI: myocardial infarction.

## Data Availability

No new data were created or analyzed in this study. Data sharing is not applicable to this article.

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
