# Peer review of "Comparison of Transcatheter Aortic Valve Implantation Devices in Aortic Stenosis: A Network Meta-Analysis of 42,105 Patients"

_jcm, 2022, doi:10.3390/jcm11185299_

Round 1

Reviewer 1 Report

Introduction:

-Please cite the 2020 AHA/ACC valve guidelines.

-Please mention the big five complications of TAVR and how do they compare with SAVR. This might improve the introduction to the study topic.

https://www.jacc.org/doi/10.1016/j.jcin.2018.12.019#:~:text=Nonetheless%2C%20we%20have%20learned%20over,)%20(10)%3B%20and%20conduction

Results:

in the study flow chart, can you also elaborate what databases were searched. Also, what were the reasons for exclusions of different studies.

Do we have baseline characteristics of cerebral embolic protection device use in any of the included studies?

For the rankogram, I would suggest using bar charts, as this will help better visualize different treatments from best to worse.

Discussion:

I feel there is room for improvement in the discussion. For a comprehensive meta-analysis, a thorough discussion is required.

With respect to PPM after TAVR please consider citing the most recent meta-analysis that provides a subgroup analysis of the different valves:

https://www.ahajournals.org/doi/10.1161/JAHA.121.020906

Also please cite prior studies that might suggest mortality benefit of TAVI compared with SAVR.

The authors mentioned that results restricted to RCTs only yielded similar results? Does that mean the results were still statistically significant? Please explain?

With respect to discussion on reduction of stroke after TAVR. Please consider discussing how, use of cerebral embolic protection devise, might have helped improved outcomes.  Consider citing the following:

https://pubmed.ncbi.nlm.nih.gov/35124076/

Please cite studies that support use of high-deployment technique to help reduce PPM implantation after TAVR. For instance, the following study from Cleveland Clinic group might be helpful.

https://www.ahajournals.org/doi/10.1161/CIRCINTERVENTIONS.120.009407

References: The references seem to be misplaced/. Please see reference 16 to 21. They have been numbered more than once with authors name misplaced. 

Author Response

-Please cite the 2020 AHA/ACC valve guidelines.

We added the AHA/ACC guidelines (ref 11)

-Please mention the big five complications of TAVR and how do they compare with SAVR. This might improve the introduction to the study topic. https://www.jacc.org/doi/10.1016/j.jcin.2018.12.019#:~:text=Nonetheless %2C%20we%20have%20learned%20over,)%20(10)%3B%20and%20conduction

We thank the reviewer for the comment and revised the introduction accordingly and added the reference mentioned (number 14).

"Regarding this matter, the choice is still controversial, because there is scarce data comparing the different transcatheter heart valves in terms of survival and quality of life and with regards to significant complications including stroke, paravalvular leak, bleeding, acute kidney injury (AKI) and conduction abnormalities (14)." Page 2, 1st paragraph

Results:

in the study flow chart, can you also elaborate what databases were searched. Also, what were the reasons for exclusions of different studies.

We thank the reviewer for the comments and added the databases searched and exclusion criteria.

"Devices that are not commercially available, were not included in the current analysis, studies without outcome report or lack of matching were not included as well. Three independent investigators (AA and AAD and AEL) had systematically screened (August 2020) MEDLINE/PubMed/Ovid/Embase for titles and abstracts containing the terms “TAVI” OR “TAVR” OR “Aortic stenosis”, reviewed the full-text articles and determined their eligibility" Page 2, Methods section

Do we have baseline characteristics of cerebral embolic protection device use in any of the included studies?

Embolic protection use became available recently and therefore we believe its use was limited in the included studies, moreover in RCTs evaluating a certain valve embolic protection was not included.

We added this point to the discussion section.

"Moreover, the development of embolic protection devices and improvement of implantation techniques to reduce pacemaker rate will further advance TAVI procedure to become safer with less (44, 45)" Page 14, 2nd paragrpah

For the rankogram, I would suggest using bar charts, as this will help better visualize different treatments from best to worse.

We thank the reviewer for the comment, however we use the surface under the cumulative ranking (SUCRA), hoping it is clearer.

Discussion:

I feel there is room for improvement in the discussion. For a comprehensive meta-analysis, a thorough discussion is required.

With respect to PPM after TAVR please consider citing the most recent meta-analysis that provides a subgroup analysis of the different valves:

https://www.ahajournals.org/doi/10.1161/JAHA.121.020906

Also please cite prior studies that might suggest mortality benefit of TAVI compared with SAVR.

We thank the reviewer for the comment and added the suggested reference (number 38).

The authors mentioned that results restricted to RCTs only yielded similar results? Does that mean the results were still statistically significant? Please explain?

We thank the reviewer for the comment, and added a clarification of the statement along with a description of the results in the supplement sections (Figures S1-S2).

"When limiting the analysis to RCTs, there was no statistically significant difference in all-cause mortality, stroke and aortic regurgitation between the various devices and SAVR, however the overall trend and ranking analysis yielded similar results. (Figure 6). While heterogeneity and quality differences do exist between studies in specific outcomes comparisons (Supplementary Tables 1-3, Figures 1 and 2), inconsistency and node-split analysis showed also produced similar results." Page 10, last paragraph.

With respect to discussion on reduction of stroke after TAVR. Please consider discussing how, use of cerebral embolic protection devise, might have helped improved outcomes.  Consider citing the following:

https://pubmed.ncbi.nlm.nih.gov/35124076/

Please cite studies that support use of high-deployment technique to help reduce PPM implantation after TAVR. For instance, the following study from Cleveland Clinic group might be helpful.

https://www.ahajournals.org/doi/10.1161/CIRCINTERVENTIONS.120.009407

We thank the reviewer for the comments and added that to the discussion section:

"Moreover, the development of embolic protection devices and improvement of implantation techniques to reduce pacemaker rate will further advance TAVI procedure to become safer with less (44, 45)"

References: The references seem to be misplaced/. Please see reference 16 to 21. They have been numbered more than once with authors name misplaced. 

We thank the reviewer for the comments and revised the section.

Reviewer 2 Report

The authors may improve their manuscript if they are willing to proceed with the following :

1. Include only RCTs 

2. Provide detailed Table regarding methods used in the included studies in order to define the main study outcome events

3. Provide measures of quality of the included studies, as well as heterogeneity

4. Provide details regarding the use of annular vs supra-annular devices relative to aortic root anatomy

5. Provide details regarding coronary alignment

6. Re-write the ranking analysis in order the average reader to comprehend on the method used to rank the different devices 

Author Response

We Thank the reviewer for the comments and have made the following changes:

1. Include only RCTs 

We included a separate analysis of RCTs only, showing similar trends.

When limiting the analysis to RCTs, there was no statistically significant difference in all-cause mortality, stroke and aortic regurgitation between the various devices and SAVR, however the overall trend and ranking analysis yielded similar results. (Figure 6). While heterogeneity and quality differences do exist between studies in specific outcomes comparisons (Supplementary Tables 1-3, Figures 1 and 2), inconsistency and node-split analysis showed also produced similar results. (page 10, last paragraph).

(figure 6, Figure S1, S2)

2. Provide detailed Table regarding methods used in the included studies in order to define the main study outcome events

The study design is reported in table 1 as RCT or propensity matched.

3. Provide measures of quality of the included studies, as well as heterogeneity

We included a quality analysis and heterogeneity assessment in the supplements (Tables S1-3).

4. Provide details regarding the use of annular vs supra-annular devices relative to aortic root anatomy

The comparison is between the different valve designs and generation, as such Sapien and Portico are annular while Acurate and Evolut are supra-annular, however we do not have specific details of outcome according to the aortic root anatomy and therefore are limited in such analysis.

5. Provide details regarding coronary alignment

Coronary alignment techniques were introduced in recent years while most studies are older and we believe that these techniques were not used as they were not reported in the methods of the different studies.

We included this in the limitations section:

"Moreover, the development of embolic protection devices and improvement of implantation techniques to reduce pacemaker rate will further advance TAVI procedure to become safer with less (44, 45)" page 14, 2nd paragraph.

6. Re-write the ranking analysis in order the average reader to comprehend on the method used to rank the different devices 

We thank the reviewer for the comment and revised the manuscript accordingly.

"Rankings according to the probability of being the best devices among the various TAVI devices and SAVR based on the Bayesian network meta-analysis revealed that Sapien 3 was ranked as having the best probability for being the most effective valve in reduction of all-cause mortality, cardiovascular mortality, stroke and bleeding, while Evolut R/Evolut Pro was ranked together with Sapien 3 with regards to atrial fibrillation, rehospitalization and AKI as shown in Figure 5. SAVR and Sapien devices were ranked highest probability for decreased incidence of pacemaker implantation. Sapien 3 and SAVR were ranked as having the highest probability for decreasing the risk of aortic regurgitation." Page 10, 1st paragraph.
